# Isolation and Characterization of a Novel *Vibrio natriegens*—Infecting Phage and Its Potential Therapeutic Application in Abalone Aquaculture

**DOI:** 10.3390/biology11111670

**Published:** 2022-11-17

**Authors:** Xuejing Li, Yantao Liang, Zhenhua Wang, Yanyan Yao, Xiaoli Chen, Anran Shao, Longfei Lu, Hongyue Dang

**Affiliations:** 1State Key Laboratory of Marine Environmental Science, Fujian Key Laboratory of Marine Carbon Sequestration, College of Ocean and Earth Sciences, Xiamen University, Xiamen 361102, China; 2College of Marine Life Sciences, Institute of Evolution and Marine Biodiversity, Frontiers Science Center for Deep Ocean Multispheres and Earth System, Ocean University of China, Qingdao 266003, China; 3Weihai Changqing Ocean Science Technology Co., Ltd., Weihai 264316, China

**Keywords:** bacteriophage, phage therapy, alternative antimicrobial, phage genome, abalone, marine aquaculture

## Abstract

**Simple Summary:**

*Vibrio* bacteria are ubiquitous and abundant in coastal waters and sediments, and some species are pathogens to humans and marine organisms, leading to foodborne diseases and pandemic infections in humans and mass mortality and economic losses in marine aquaculture. However, the misuse and overuse of antibiotics have led to the emergence and spread of antibiotic-resistant bacteria and resistance genes in marine aquacultural settings and adjacent coastal environments. Bacterial pathogens may also acquire antibiotic resistance via gene mutations and horizontal gene transfers in such environments, making bacterial infectious diseases extremely difficult or even impossible to treat using conventional antibiotic-based therapies. Consequently, phage therapy has received increasing attention in recent years. This study describes the characterization of phage vB_VnaS-L3 from *Vibrio natriegens*, and explores its potential application as a biocontrol agent to replace antibiotics against pathogenic *V. natriegens* in juvenile Pacific abalones aquaculture. Our findings showed that phage vB_VnaS-L3 could be a potential alternate biocontrol and prophylactic agent that could effectively inhibit the growth of pathogenic *V. natriegens*, significantly reduce the mortality of juvenile abalones, and maintain abalone feeding capacity.

**Abstract:**

Phage-based pathogen control (i.e., phage therapy) has received increasing scientific attention to reduce and prevent the emergence, transmission, and detrimental effects of antibiotic resistance. In the current study, multidrug-resistant *Vibrio natriegens* strain AbY-1805 was isolated and tentatively identified as a pathogen causing the death of juvenile Pacific abalones (*Haliotis discus hannai* Ino). In order to apply phage therapy, instead of antibiotics, to treat and control *V. natriegens* infections in marine aquaculture environments, a lytic phage, vB_VnaS-L3, was isolated. It could effectively infect *V. natriegens* AbY-1805 with a short latent period (40 min) and high burst size (~890 PFU/cell). Treatment with vB_VnaS-L3 significantly reduced the mortality of juvenile abalones and maintained abalone feeding capacity over a 40-day *V. natriegens* challenge experiment. Comparative genomic and phylogenetic analyses suggested that vB_VnaS-L3 was a novel marine *Siphoviridae*-family phage. Furthermore, vB_VnaS-L3 had a narrow host range, possibly specific to the pathogenic *V. natriegens* strains. It also exhibited viability at a wide range of pH, temperature, and salinity. The short latent period, large burst size, high host specificity, and broad environmental adaptation suggest that phage vB_VnaS-L3 could potentially be developed as an alternative antimicrobial for the control and prevention of marine animal infections caused by pathogenic *V. natriegens*.

## 1. Introduction

Bacteria affiliated with the *Vibrio* genus are ubiquitous and abundant in coastal waters and sediments [1]. These bacteria usually exhibit a multiplicity of lifestyles, which play an essential role in their physiological plasticity and versatile ecological functioning [2,3]. For example, the association with marine phytoplankton, zooplankton, organic particles, microplastics, and other solid surfaces helps vibrios access extra nutrients and organic substrates and provides a mechanism for the vibrios to cope with adverse conditions [4]. Surface colonization also contributes critically to establishing close interactions of vibrios with their host organisms in a mutualistic, commensalistic, or parasitic relationship [5]. Furthermore, surface colonization, which usually leads to biofilm formation, protects vibrios against protist predation, antibiotics, biocides, and other harmful agents [4]. Some *Vibrio* species are pathogens to humans and marine organisms, leading to foodborne diseases and pandemic infections in humans [2] and mass mortality and economic losses in marine aquaculture [6]. Aquaculture usually uses antibiotics to control and prevent vibriosis and other bacterial diseases [7]. However, the misuse and overuse of antibiotics have led to the emergence and spread of antibiotic-resistant bacteria (including many species of *Vibrio*) and resistance genes in marine aquacultural settings and adjacent coastal environments [8], which also receive other anthropogenic sources of antibiotic pollution and antibiotic-resistant bacterial contamination [9,10]. Bacterial pathogens may acquire antibiotic resistance via gene mutations and horizontal gene transfers in such environments, making bacterial infectious diseases extremely difficult or even impossible to treat using conventional antibiotic-based therapies [11].

Antibiotic resistance and its spread pose a growing global health crisis and threat, predicted to cause 10 million deaths per year after 2050 [12]. As hotspots for antibiotic pollution and the development and dissemination of antibiotic-resistant bacteria and resistance genes, aquaculture-influenced environments have received increasing attention amid concerns about food safety, public health, and ecosystem sustainability [11]. Developing alternative therapeutics instead of antibiotics is needed to mitigate antibiotic resistance in aquacultural pathogen control and prevention [13]. Bacteriophages (phages, for short) are viruses that specifically infect bacteria. Viruses are the most abundant and diverse biological entities in the ocean, playing a pivotal role in regulating microbial abundance, community structure, and biogeochemical cycling of various bioessential elements [14]. Phages have been proposed as promising alternatives to conventional antibiotics for treating bacterial infectious diseases, particularly those caused by antibiotic-resistant pathogens [15]. Phages are highly host-specific and non-toxic toward higher organisms, posing no threat to humans, animals, and plants [16]. Many phages encode genes for the enzymes that specifically target and destroy bacterial biofilms, that may render bacteria 1000 times more resistant to antibiotics [16]. In addition, phage therapy is deemed an ecologically safe and environmentally-friendly treatment, having minimal impacts on the environment and the microbial community except for the targeted host bacteria [16]. These advantages make phage therapy particularly suitable for the treatment of pathogenic bacteria in aquaculture [17]. Furthermore, as “live drugs” (self-replicating in response to host abundance), phages possess inherent auto-dosing and augmentation effects during treatment, providing an economical and convenient way for pathogen control in large aquacultural water bodies [18].

Many phages can infect vibrios. Approximately 850 *Vibrio*-infecting phages have been isolated, and their genomic sequences have been deposited in the National Center for Biotechnology Information (NCBI) database so far (as of 3 October 2022). The hosts of these phages mainly belong to *V. cholerae*, *V. parahaemolyticus*, and *V. vulnificus* [19]. Many *Vibrio*-infecting phages show biocontrol potentials because of their ability to suppress host growth and disrupt biofilms [20,21]. Although the arms race between bacteria and phages makes it unavoidable for phage resistance to occur in host bacteria, bacterial resistance to phages occurs at much lower rates than the rates of bacterial resistance to antibiotics [22]. Moreover, the development of phage resistance may sensitize the host bacteria to antibiotics or make the host bacteria less virulent, rendering the phage steering effects exploitable for bacterial pathogen control [23]. Therefore, phage control is a reasonable option to treat and prevent the establishment of *Vibrio* pathogens in marine aquaculture. In the current study, three marine bacterial strains, including AbY-1805, AbY-1806, and AbY-1807, were isolated and identified as pathogenic *Vibrio natriegens* that could cause the death of juvenile abalones (*Haliotis discus hannai* Ino). A *V. natriegens* AbY-1805-infecting phage, vB_VnaS-L3, was further isolated. We studied the genome of vB_VnaS-L3 and explored the potential to use this marine phage as a biocontrol agent for treating and preventing abalone infections caused by pathogenic *V. natriegens*.

## 2. Materials and Methods

### 2.1. Bacterial Strains and Culture Conditions

Three bacterial strains, including AbY-1805, AbY-1806, and AbY-1807, were isolated from the tissue of diseased abalone (*Haliotis discus hannai* Ino) and the rearing water in an abalone hatchery in Sanggou Bay, China, in 2018. These bacteria were characterized via 16S rRNA gene sequence analyses, and AbY-1805 was further selected to test its pathogenicity toward juvenile abalones. In addition, strain AbY-1805 was used as a host for phage isolation, in order to obtain marine phage vB_VnaS-L3. A total of 39 bacterial strains (Appendix A) were used to investigate the host range of phage vB_VnaS-L3. All the bacteria were cultivated at 28 °C in RO medium, an artificial seawater medium containing 1 g/L yeast extract, 1 g/L tryptone, and 1 g/L sodium acetate at pH 7.5 to the log phase (OD 0.5–0.6). The 16S rRNA gene sequences of strain AbY-1805 and other bacteria (excluding *V. natriegens* strains MCCC 1A14388, MCCC 1K03861, and MCCC 1H000251 that were purchased from the Marine Culture Collection of China, Xiamen, China) used for host range investigation were submitted to GenBank under accession numbers OP247638 and OP268245-OP268279.

Antibiotic susceptibility tests revealed that strain AbY-1805 was resistant to many (17 out of 32) antibiotics (Appendix A). Oxytetracycline, a commonly used antibiotic in aquaculture, showed good efficacy against strain AbY-1805. This antibiotic was used in the subsequent bioassay experiment to compare the antibiotic therapy effect with the effect of phage therapy using vB_VnaS-L3 against strain AbY-1805.

### 2.2. Phage Isolation and Characterization

Seawater samples were collected from a marine aquaculture area in Sanggou Bay, China, in December 2018 and filtered through 0.22 μm filter membranes (Millipore, Bedford, MA, USA), immediately. The filtrate was added to a logarithmic-phase culture of strain AbY-1805, incubated at 28 °C for 24 h, 36 h, and 48 h, respectively, and then phages were detected by the double-agar layer method [24]. The observed single plaques were picked with 1000-μL sterilized wide-bore pipette tips and plaque-purified five times, as described in a previous study [25]. Then the purified phages were stored in sodium chloride-magnesium sulfate (SM) buffer (100 mM NaCl, 50 mM Tris, 10 mM MgSO_4_, and 0.01% gelatin, pH 7.5) with chloroform at −80 °C. The obtained phage was named vB_VnaS-L3.

The host range of phage vB_VnaS-L3 was determined with 39 bacterial strains using a spot test by adding 5 μL of a diluted phage suspension (~10^8^ PFU/mL) dropwise onto the surface of double-layer agar plates inoculated with a tested bacterial strain. The plates were incubated at 28 °C for up to 7 days, and plaque formation was assessed repeatedly during this period. The tested bacterial strains included twenty-five *Vibrio* species, five *Shewanella* species, two *Photobacterium* species, two *Pseudoalteromonas* species, one *Idiomarina* species, and one *Sulfitobacter* species (Appendix A). Positive bacterial strains were further tested by using different phage dilutions (~10^6^, ~10^7^ and ~10^8^ PFU/mL) to confirm the results.

The morphology of phage vB_VnaS-L3 was determined using a 100 kV TEM (JEOL Model JEM-1200EX, JEOL, Tokyo, Japan), and images were taken using the GATAN INC CCD image transmission system (Gatan, Pleasanton, CA, USA).

One-step growth curves were determined as previously described [25]. In brief, 100 mL bacterial strain AbY-1805 cells (OD_600_ of 0.5–0.6) was mixed with phage vB_VnasS-L3 at a multiplicity of infection (MOI) of 0.01. After adsorption for 10 min at 28 °C, the mixtures were centrifuged at 6000× *g* for 10 min to remove non-adsorbed phage particles. Then the pelleted cells were resuspended in 100 mL RO medium, and incubated at 28 °C. Aliquots were collected every 10 min over a 60-min period, and phage titers were determined by the double-layer plaque assay method [24]. The optical density (OD_600_) of samples were measured at the same time. Three independent experiments were performed.

The lysis assay for phage vB_VnaS-L3 on bacterial strain AbY-1805 was determined at various MOIs (0.001, 0.01, 0.1, 1). Briefly, 100 mL bacterial strain AbY-1805 cells were grown to an OD_600_ = 0.5–0.6, and then mixed with phage vB_VnasS-L3 at different MOI, respectively. Sample OD_600_ values were measured every 15 min to monitor the growth condition of the bacterial strain AbY-1805 over 300 min, and the results were further confirmed by CFU counts at 0, 150, and 300 min. The titer of each initial phage inoculum was determined by the double-layer plaque assay method [24]. Triplicate determinations were performed for each assay.

### 2.3. Phage Stability Characterization

Phage stability under different pH, temperature, and salinity conditions was tested separately. For the phage stability experiment under different pH, the pH of the SM buffer was adjusted to 6–10 with 5 M HCl or NaOH solution. After being filtered using a 0.22 μm membrane filter (Millipore, Bedford, MA, USA), each 9 mL SM buffer was added to 1 mL phage suspension with a titer value ~10^7^ PFU/mL, incubated at 28 °C for 24 h, and then the infection activity of the phage was determined by the double-layer agar method. The phage stability under different temperatures was examined by storing 1 mL of phages (~10^7^ PFU/mL) at 4 °C to 40 °C for 24 h, then cooled to 4 °C immediately for phage infection activity examination by the double-agar layer method. The phage stability under different salinities was analyzed by aliquoting 1 mL of phages in centrifuge tubes with varying percentages (0% to 100%) of sterile seawater (34‰) adjusted with sterile freshwater with the same phage incubation and infection activity determination procedures as those used in the phage pH stability experiment.

### 2.4. Phage DNA Extraction, Genome Sequencing, and Genome Assembly

Phage genomic DNA was extracted using the phenol/chloroform DNA extraction method [26]. In order to avoid contaminations from bacterial materials, bacterial nucleic acids were removed through digestion with DNase I and RNase A (Takara Bio Inc., Shiga, Japan) at 37 °C for 1 h prior to phage DNA extraction. Then the solution was incubated with sodium dodecyl sulfate (final concentration 1% *w*/*v*) and proteinase K (final concentration 100 μg/mL) at 55 °C for 2 h [26]. Phage genome sequencing was performed using the Illumina NovaSeq 6000 platform by the Shanghai Biozeron Biotechnology Co., Ltd. (Shanghai, China).

High-quality clean sequence data were obtained using Trimmomatic (version 0.36) with parameters (SLIDINGWINDOW: 4:15, MINLEN: 75) [27]. Then the phage genome was assembled using ABySS with multiple-Kmer parameters [28]. Finally, the GapCloser software (San Francisco, CA, USA) was applied to fill the remaining local inner gaps and correct the single base polymorphism for the final assembly results [29]. The complete genome sequence of phage vB_VnaS-L3 was submitted to the GenBank database (accession number ON714422).

### 2.5. Phage Genomic and Phylogenetic Analyses

GeneMarkS online server (http://exon.gatech.edu/Genemark/genemarks.cgi, accessed on 10 May 2020) [30], Glimmer 3.0 (http://ccb.jhu.edu/software/glimmer/index.shtml, accessed on 10 May 2020) [31], and ORF Finder online server (https://www.ncbi.nlm.nih.gov/orffinder/, accessed on 10 May 2020) were used for the prediction of putative open reading frames (ORFs) of the assembled genome of phage vB_VnaS-L3. The NCBI non-redundant (NR) database, SwissProt (http://uniprot.org, accessed on 10 May 2020), KEGG (http://www.genome.jp/kegg/, accessed on 10 May 2020), and COG (http://www.ncbi.nlm.nih.gov/COG, accessed on 10 May 2020) were used for functional annotation of the predicted ORFs. In addition, tRNAs were identified using tRNAscan-SE (v1.23, http://lowelab.ucsc.edu/tRNAscan-SE, accessed on 10 May 2020) [32,33]. OrthoFinder was used to compare the genomic similarity between phage vB_VnaS-L3 and its closest neighbors, and the average nucleotide identity by orthology (OrthoANI) was calculated using the all-vs-all BLASTp analysis [34,35]. A phage proteomic tree was constructed using the Viral Proteomic Tree server (ViPTree, https://www.genome.jp/viptree/, accessed on 13 May 2022) based on genome-wide sequence similarities computed by tBLASTx to determine the taxonomy of the isolated phage vB_VnaS-L3 [36]. In addition, the phage major capsid protein sequences were used to construct the maximum-likelihood phylogenetic tree using MEGA X with the “WAG + G” model, with reference sequences of closely related phages and other phages infecting *V. natriegens* being retrieved from GenBank.

### 2.6. Phage Biocontrol Potential Bioassay

Seven hundred and twenty juvenile abalones of 27.62 ± 1.48 mm shell length were equally divided into four groups (each containing three replicates) for the bioassay. Before the pathogen challenge started, these abalones went through an acclimation stage by being maintained in aquaria at 12 °C with aerated running seawater for 6 days and fed with fresh kelp, which was also used as the feed for the abalones during the subsequent course (immunization, inoculation, and observation stages) of the bioassay (Table 1). All four groups received live pathogenic AbY-1805 cells (at a final concentration of 10^5^ cells/mL) every day during the inoculation stage. In addition, heat-inactivated AbY-1805 cells (at a final concentration of 10^5^ cells/mL) were added every day to the immunity-preboosting group during the immunization stage, 10 ppm oxytetracycline was added every day to the antibiotic-treatment group during the inoculation stage, and phage vB_VnaS-L3 suspension (at a final concentration of 10^6^ CFU/mL) was added every day to the phage-treatment group during both immunization and inoculation stages. The live pathogen-only group that received only live AbY-1805 cells during the inoculation stage was used as the control group in the experiment design. The phage suspension of vB_VnaS-L3 was obtained from 0.22 μm-filtered phage lysate in incubation with bacterial strain AbY-1805, and phage titer was determined by the double-layer agar method. During the immunization stage, heat-inactivated AbY-1805 suspension was added to the immunity-preboosting group to activate the innate immunity of abalone since the antigen epitopes could be retained during heat inactivation of the pathogen [37]. Since many studies have confirmed the preventive effect of prophylactic phage administration, phage vB_VnaS-L3 suspension was added to the phage-treatment group during the immunization stage [38,39,40,41]. After the inoculation stage, all four groups were kept to continue for another 10 d (observation stage) without any addition of tested bacterial strain, phage, or oxytetracycline to any of the groups [42]. The abalone feeding rate was calculated daily by measuring the reduced weight of fresh kelp (as feed).

## 3. Results

### 3.1. Isolation and Phylogenetic Characterization of the Pathogenetic Bacterial Strain AbY-1805

Three bacterial pathogens were isolated in this study that caused 50–80% mortality of juvenile abalones at 24 °C and >40% decrease in feeding rates of Yesso scallop (*Mizuhopecten yessoensis*) and Zhikong scallop (*Azumapecten farreri*) at 8–12 °C. Phylogenetic analyses based on their 16S rRNA gene sequences (Figure 1, Appendix A) showed that all these three bacteria were affiliated with Vibrio natriegens. Strain AbY-1805 was used for subsequent experiments in this study. The antibiotic susceptibility tests of strain AbY-1805 showed that this bacterium was resistant to more than half of the tested antibiotics (Appendix A).

### 3.2. Isolation and Characterization of Phage vB_VnaS-L3

A phage, designated as vB_VnaS-L3, was isolated from aquaculture seawater in the Sanggou Bay using *V. natriegens* AbY-1805 as a host. Small, clear, and round plaques were observed on the lawn of *V. natriegens* (Figure 2A). Morphological analysis showed that phage vB_VnaS-L3 belongs to family *Siphoviridae*, with an icosahedral capsid (57.4 ± 3.3 nm) and a long-sheathed tail (160.0 ± 20.2 nm) (Figure 2B).

All 39 bacterial strains used in the test of the host range of phage vB_VnaS-L3 belonged to Gram-negative bacteria (Appendix A). Phage vB_VnaS-L3 could only infect three strains of *V. natriegens*, showing a narrow host range and the potential as a biocontrol agent specifically targeting only certain species of *V. natriegens*. The one-step growth curve and infective activity of phage vB_VnaS-L3 against *V. natriegens* strain AbY-1805 were further determined (Figure 3). The latent period for phage vB_VnaS-L3 was about 40 min, and the burst size was approximately 890 ± 25 PFU/cell. High infective activity of phage vB_VnaS-L3 was observed, and the growth of *V. natriegens* strain AbY-1805 was inhibited even at the lowest (0.001) multiplicity of infection (MOI) (Figure 3C,D).

To explore the stability of phage vB_VnaS-L3 under different environmental conditions, phage titers were determined after phage vB_VnaS-L3 was incubated for 24 h at different pH, temperature, and salinity, respectively. Phage vB_VnaS-L3 showed a broad pH stability (pH 6 to 10), with the highest titer being observed at pH 7.5 (Figure 4A). Notably, more than 40% of the phages still survived even at pH 6 or pH 10. The thermal stability of phage vB_VnaS-L3 is shown in Figure 4B. In general, the phage titer decreased with increasing temperature (from 4 °C to 40 °C), and phage vB_VnaS-L3 was still moderately stable at 40 °C with a survival rate of about 68%. Phage vB_VnaS-L3 could keep high activity level (>88%) with freshwater addition ratios under 20%. However, the phage activities decreased sharply when the freshwater addition ratios exceeded 20% (Figure 4C).

### 3.3. Genomic Characterization of Phage vB_VnaS-L3

Illumina NovaSeq 6000 platform was used for sequencing the phage vB_VnaS-L3 genomic DNA, resulting in a library of 2582.2 Mb clean sequence data, with an average sequence length of 150 bp. A circular 39,988-bp double-stranded DNA genome was obtained after de novo assembly, and the G + C content was 43.74%. The total number of predicted open reading frames (ORFs) was 61, comprising 91.1% of the genome (Appendix A). In total, 56 ORFs were identified to share sequence homology with known bacteriophage genes, including 14 ORFs that mainly encoded for gene fragments related to phage structure and function, with the similarity ranging from 41.0% to 98.7% with known phage genes (Appendix A and Figure 5). Three predicted ORFs were found to encode structural components of the phage, including ORF 3 (encoding tail tape measure protein), ORF 54 (encoding major capsid protein), and ORF 58 (encoding head-closure protein). Ten predicted ORFs encoded genes involved in DNA metabolism, including two helicase-encoding ORFs (ORF 24 and ORF 29), two DNA polymerase-encoding ORFs (ORF 31 and ORF 32), one DNA-binding domain protein-encoding ORF (ORF 38), one DNA-packaging protein-encoding ORF (ORF 41), one terminase large subunit-encoding ORF (ORF 42), one NAD-asparagine ribosyltransferase-encoding ORF (ORF 48), and two DNA methylase-encoding ORFs (ORF 52 and ORF 57). In addition, a host lysis-related ORF (ORF 14) was recognized in the genome of phage vB_VnaS-L3.

### 3.4. Phylogenetic and Comparative Genomic Analyses of Phage vB_VnaS-L3

A genome-based proteomic tree was constructed using the ViPTree server to determine the taxonomic affiliation of phage vB_VnaS-L3 (Figure 6A,B). In this tree, phage vB_VnaS-L3 was clustered with *Listonella pelagia* phage φHSIC (Genbank accession no. NC_006953) and *Vibrio* sp. YD38 phage pYD38-B (NC_021561) within the viral family *Siphoviridae*, consistent with the morphological observation of phage vB_VnaS-L3 (Figure 1B). The proteomic tree (Figure 6A,B) also indicated that the host group of phage vB_VnaS-L3 belonged to *Gammaproteobacteria*. Based on the phylogenetic tree constructed using protein sequences of the viral major capsid protein (MCP), phage vB_VnaS-L3 was clustered with Vibrio phage ValSw3-3 (MG676223) and it also showed close evolutionary relationships with Listonella phage φHSIC, Vibrio phage P23 (MK097141), and Vibrio phage pYD38-B (Figure 6C). The values of orthologous average nucleotide identity (OrthoANI) calculated from complete genome sequences indicated the closest association of vB_VnaS-L3 with Vibrio phage pYD38-B (OrthoANI value = 84.81%) (Figure 6D).

Genome-wide comparisons between phage vB_VnaS-L3 and its most closely related phages, Listonella phage φHSIC and Vibrio phage pYD38-B, showed substantial functional differences between phage vB_VnaS-L3 and phage φHSIC, whereas similar functional domains could be found between phage vB_VnaS-L3 and Vibrio phage pYD38-B. Only phage vB_VnaS-L3 contained DNA methylase-encoding ORFs, while a putative hemagglutinin protein-encoded ORF was found only in the genome of phage φHSIC (Figure 7). Moreover, ORF homology comparisons between phage vB_VnaS-L3 and the two closely related phages were carried out. Thirty and 28 ORFs of phage vB_VnaS-L3 were found to have 56.4–98.7% and 59.0–90.8% homology with Listonella phage φHSIC and Vibrio phage pYD38-B, respectively. All the above analytical results indicated that phage vB_VnaS-L3 represented a novel *Siphoviridae* phage specifically infecting *V. natriegens*.

### 3.5. Bacterial Pathogen Biocontrol Potential of Phage vB_VnaS-L3

Throughout the bioassay experiment (including the acclimation, immunization, inoculation, and observation stages), the phage-treatment group showed the highest abalone survival rate (95.0% on average), followed by the antibiotic-treatment group (90.5% on average), immunity-preboosting group (84.4% on average), and live pathogen-only group (62.2% on average) (Figure 8A).

Abalone feeding rates were measured as an indicator of abalone physiological status during the bioassay. No significant differences in abalone feeding rate were found in the acclimation and immunization stages among the four groups (except for day 14). In contrast, significantly higher abalone feeding rates were observed for the phage-treatment group than for the other three groups (except for the antibiotic-treatment group on day 26) at late time points of the inoculation and observation stages. In addition, the live pathogen-only group usually had the lowest abalone feeding rates during the inoculation and observation stages (Figure 8B). The difference in the abalone physiological statuses, as suggested by the abalone feeding rates, among the distinct groups indicated that the designed treatments in the immunity-preboosting group (immunostimulating therapy), antibiotic-treatment group (antibiotic chemotherapy), and phage-treatment group (phagotherapy) indeed had certain but varying degrees of biocontrol effects on the abalone pathogen, *V. natriegens* strain AbY-1805. This result is highly consistent with the observed abalone survival rates (Figure 8A). Our experiments indicated that pathogen biocontrol using phage vB_VnaS-L3 was an effective and promising treatment for *V. natriegens* infections in juvenile abalone.

## 4. Discussion

The global decline in wild fish catches, growth in seafood consumption, expansion in international trade and domestic markets, escalation in urbanization, and rise in personal incomes are among the leading contributors driving the rapid development of aquaculture worldwide in the past decades [43]. As a result, world food production via aquaculture has already surpassed that from wild fishery harvests [44]. However, the sustainability of marine aquaculture confronts serious challenges such as pathogens, harmful algal blooms, and climate change-induced environmental alternation in the ocean (e.g., deoxygenation and acidification) [43]. Aquaculture may, in turn, negatively impact the environment and public health, including through the dissemination of pathogens and antibiotic-resistant bacteria [45]. In the current study, we isolated and identified a pathogenic strain of *V. natriegens* that could cause the massive death of juvenile abalone. In line with our finding, several other strains of *V. natriegens* have previously been identified as pathogens to other cultured marine animals, such as bay scallop (*Argopecten irradians* Lamarck) [46], Asiatic hard clam (*Meretrix meretrix*) [47], white shrimp (*Litopenaeus vannamei*) [48], kuruma prawn (*Marsupenaeus japonicus*) [49], and gazami crab (*Portunus trituberculatus*) [50]. In addition, some *V. natriegens* strains have been identified as potentially pathogenic bacteria to crown-of-thorns starfish (*Acanthasther planci*) [51].

*V. natriegens* was considered a safe bacterium (belonging to biosafety level 1) [52]. It has the fastest growth rate of all bacteria isolated so far [53]. In addition, *V. natriegens* is a facultative anaerobe well adapted to both oxic and anoxic environments [54,55]. Thus, *V. natriegens* has been explored for various bioengineering and biotechnological applications [52,54], such as serving as a whole-cell catalysis chassis for the production of L-DOPA (3,4-dihydroxyphenyl-L-alanine, a promising drug for Parkinson’s disease), violacein, β-carotene, melanin, and other pharmaceuticals and valuable chemicals [56,57,58,59]. *V. natriegens* has also been tested as an engineered “live antimicrobial” for the control of bacterial pathogens [60]. However, not all *V. natriegens* strains are as safe as initially thought. Some strains of *V. natriegens* are environmentally detrimental, contributing to biofouling and metal corrosion [61,62]. The capabilities of extracellular electron transfer and biofilm formation provide a mechanistic explanation for the detrimental effects of *V. natriegens* [63,64]. Biofilms also contribute to antibiotic resistance in *V. natriegens* pathogens [65,66]. The existence of pathogenic, biofouling, and metal-corroding strains in *V. natriegens* revealed the necessity to evaluate systematically the biologically- and environmentally-detrimental potentials of any new *V. natriegens* strains before they can be used for bioengineering and biotechnological applications [52].

In the current study, we also isolated phage vB_VnaS-L3, which showed the biocontrol potential towards abalone pathogen *V. natriegens* AbY-1805. Many different types of phages could infect *V. natriegens* [67]. However, only a limited number of *V. natriegens* phages have been described in sufficient biological details, including myoviruses KVP40, nt-1, nt-6, PWH3a-P1, and VH1_2019 [68], podovirus Phriendly [69], and siphoviruses vB_VnaS-AQKL99 and VH2_2019 [70]. In addition, no systematic study about their biocontrol potential toward *V. natriegens* infections has been made. The Pacific abalone is the most commercially and economically important aquaculture species of abalone in Asian countries [71,72]. However, abalone farming often suffers from severe economic losses caused by bacterial pathogen infections [73,74,75]. As the natural enemies to bacteria, phages that specifically infect bacterial pathogens hold great potential in controlling and preventing bacterial infections in abalone farming.

### 4.1. vB_VnaS-L3 as a Novel V. natriegens Phage

The morphological characteristics of phage vB_VnaS-L3 were similar to *V. natriegens* phages vB_VnaS-AQKL99, but were different from other *V. natriegens*-infecting phages including KVP40, nt-1, nt-6, and PWH3a-P1 [69,70,76,77]. The morphological and genomic characteristics indicated that phage vB_VnaS-L3 was affiliated with *Siphoviridae*. It had the closest phylogenetic relationships with Listonella phage φHSIC [78], Vibrio phage ValSw3-3 [19], Vibrio phage pYD38-B, and Vibrio phage P23 [79], all of which may belong to the same viral genus judged from their pairwise OrthoANI values (Figure 6D) [80]. Functional differences could be found between phage vB_VnaS-L3 and phage φHSIC, based on the genomic comparison, implying that they may have different survival strategies in their respective bacterial hosts (Figure 7). ORF homology analysis showed that less than half of the ORFs of phage vB_VnaS-L3 were homologous to its most closely related phages, indicating that phage vB_VnaS-L3 was a novel *V. natriegens* phage. Interestingly, phage vB_VnaS-L3 had quite large evolutionary distances from some other *V. natriegens*-infecting phages (Figure 6C); it may be common for a marine bacterium to host evolutionarily distant phages.

### 4.2. Biological Properties of Phage vB_VnaS-L3

Phages can be classified as lytic phages or lysogenetic phages. Good knowledge of phage biological properties is a prerequisite for the selection and proper application of phages in pathogen control. Lytic phages may be directly administered as live antimicrobials, while lysogenic phages may be more suitable for delivering engineered toxins or antimicrobial peptides for pathogen killing [81]. Basic knowledge about a phage’s host range, latent period, burst size, MOI, and environmental tolerance needs to be considered [82]. Phage vB_VnaS-L3 had a tentatively narrow host range, only infecting three strains of *V. natriegens* among the 39 tested bacteria. Target-specific bacterial lysis is a prominent advantage of phage therapy, producing minimal perturbations to microbial community structure and function [83,84].

Phage vB_VnaS-L3 possessed a broad pH, temperature, and salinity tolerance (Figure 4), suitable for application in a variety of environmental conditions. Many reported phage candidates for pathogen control also have high environmental tolerance. For instance, the lytic capacity of Vibrio phage ValSw3-3 was stable at pH 3 to 10 and temperatures of 4 to 50 °C [19]. Vibrio phage OMN was observed to tolerate changes in pH (5–9) and temperature (up to 50 °C) [85]. Moreover, phage vB_VnaS-L3 had a large burst size (~890 PFU/cell), which was about 10–40 times larger than other evolutionarily related phages and potential phage candidates for vibriosis therapy, showing a more effective pathogen lytic effect [19,79,85,86,87]. In addition, the killing curve showed that phage vB_VnaS-L3 is effective at reducing *V. natriegens* population at different MOI. Although higher titers of phage would result in faster OD reduction of bacteria, heavy phage loads could lead to the emergence of phage resistance in bacteria [88]. Therefore, we used the 10^6^ CFU/mL phage concentration in the subsequent biocontrol potential bioassay. Furthermore, Chen et al. isolated five *Aeromonas salmonicida*-infecting phages and found that phage-resistant bacteria began to emerge after 4 h at 0.1 to 10 MOI [89]. In contrast, phage vB_VnaS-L3 inhibited *Vibrio* recovery for at least 6 h even at the lowest MOI (0.001), showing a higher antibacterial activity and a more potent suppression to host resistance. Therefore, phage vB_VnaS-L3 showed a more promising potential for phage therapy application.

### 4.3. Potential Therapeutic Application of Phage vB_VnaS-L3 in Aquaculture

Bacterial infections of farmed animals result in economic losses in the aquaculture industry. Phage therapy provides a promising alternative to antibiotic therapy. It has been demonstrated for treating human pathogens, such as *Listeria monocytogenes*, *Klebsiella pneumonia*, and *Acinetobacter baumannii* [90,91]. It also has promising applications in aquaculture as an effective and inexpensive means of mitigating vibriosis for cultured vertebrate and invertebrate animals [92,93]. In the current study, we tested the application potential of phage therapy in abalone aquaculture upon pathogenic *V. natriegens* in a 40-day in vivo bioassay experiment.

*V. natriegens* could cause heavy mortality in shellfish aquaculture [46,48,50]. In the present study, *V. natriegens* strain AbY-1805 was found to be a multidrug-resistant bacterial pathogen that caused the massive death of juvenile abalone (Appendix A). Our experiment demonstrated that using phage vB_VnaS-L3 as a therapeutic treatment could achieve positive results against abalone vibriosis caused by *V. natriegens* AbY-1805. Compared with the antibiotic-treatment group and immunity-preboosting group, the phage-treatment group generally performed better in increasing the abalone survival and feeding rates (Figure 8). These results indicated that phage therapy using vB_VnaS-L3 tended to be effective for preventing and controlling *V. natriegens*-induced vibriosis. The efficacy of phage treatment to vibriosis diseases of greenlip abalone (*Haliotis laevigata*) has previously been demonstrated with two bacteriophages (vB_VhaS-a and vB_VhaS-tm) against *V. harvieyi* [94]. In a bioassay experiment using zebrafish larvae, directly adding phages to the zebrafish culture could significantly reduce fish mortality [95]. Similarly, in another study, the growth of *V. parahaemolyticus* in cultured shrimp could be effectively inhibited by a phage cocktail added to the culture system [96]. The consistently demonstrated successes (including our current study) in the trials of bacterial pathogen control using phages as preventive or/and therapeutic agents warrant further exploration and development of phage control techniques for sustainable aquaculture.

The addition of phage vB_VnaS-L3 or heat-inactivated *V. natriegens* AbY-1805 before the cultured juvenile abalones were challenged with live *V. natriegens* AbY-1805 offered the abalones better resistance to the bacterial pathogen (Figure 8). Exposure to heat-killed bacterial pathogen cells as immunopotentiators may upregulate the expression of the animals’ immune effector factors, as confirmed in many previous studies [97,98]. In several in vivo and in vitro experiments, phages have also been confirmed to affect the animals’ epithelial, endothelial, and immune cells, dampening the inflammatory response stimulated by bacterial pathogens [99,100,101]. A *Vibrio*-infecting phage also showed prophylactic characteristics in the challenge experiment of larval oysters in vivo, showing a significantly decreased mortality rate when the oysters were treated with phage 15 min before the pathogen challenge [38].

Phage therapy shows good prevention and treatment effects for controlling vibriosis and other bacterial infections in aquaculture. Our current study also showed that the juvenile abalones maintained quite healthy physiological status (as revealed by the abalones’ feeding rates) with the treatment of phage vB_VnaS-L3 (Figure 8), suggesting that phage therapy may be suitable for applications in full-scale culture ponds. Moving from laboratory experiments to field experiments may push phage therapy techniques to the next level for testing and in-reality applications.

## 5. Conclusions

In this study, a *V. natriegens* strain, AbY-1805, was isolated and identified as an abalone pathogen. In addition, a novel bacteriophage, vB_VnaS-L3, was successfully isolated from a marine aquaculture environment and identified as a *Siphoviridae* phage that could infect *V. natriegens*. Phage vB_VnaS-L3 had a huge burst size and broad pH, thermal, and salinity tolerance, potentially useful traits for phage therapy applications. Furthermore, phage vB_VnaS-L3 showed effective inhibition of *V. natriegens* AbY-1805 growth in vitro and in vivo, and it could also maintain juvenile abalones in relatively healthy physiological status. Therefore, phage vB_VnaS-L3 may be a good candidate for applying phage therapy and prevention against pathogenic *V. natriegens* disease outbreaks in marine aquaculture. In practice, prevention is more important than treatment in the large-scale aquacultural industry. Further studies should be performed to evaluate the efficacy of phage vB_VnaS-L3 as a therapeutic and prophylactic agent in aquaculture field applications.

## Figures and Tables

**Figure 1 biology-11-01670-f001:**
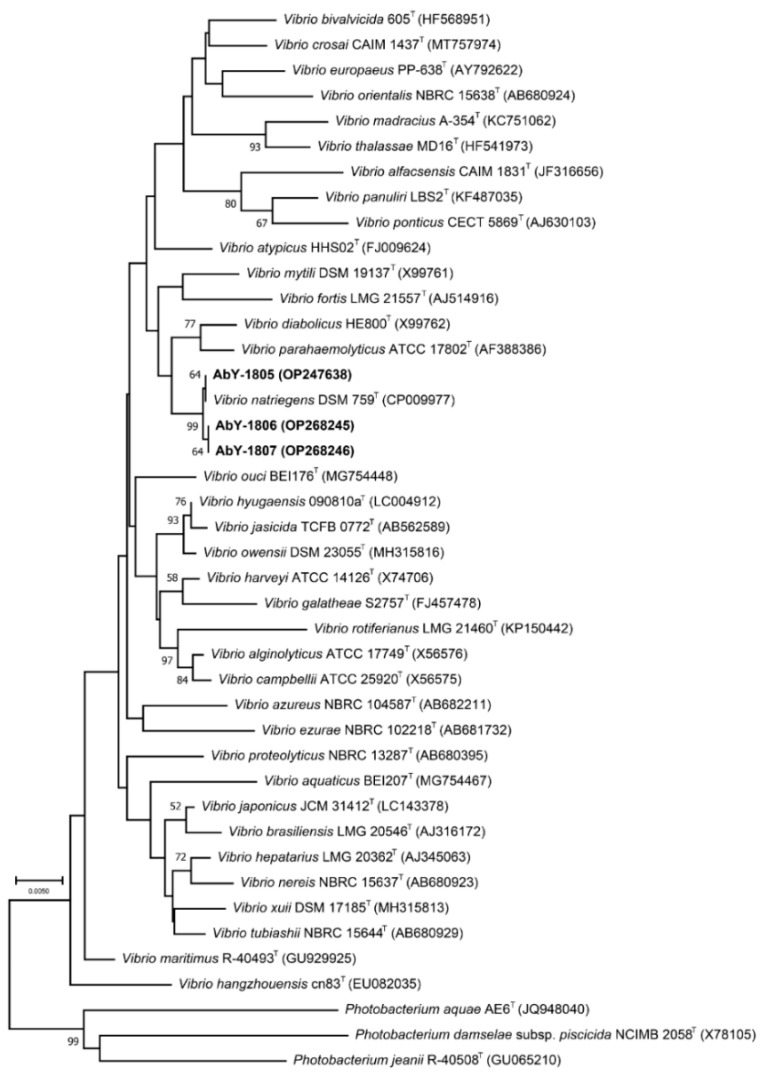
Neighbor-joining phylogenetic tree based on bacterial 16S rRNA gene sequences. Bootstrap values (1000 resampling) higher than 50% are shown near the corresponding nodes. The 16S rRNA gene sequences of the three pathogenic strains obtained in this study are shown in bold.

**Figure 2 biology-11-01670-f002:**
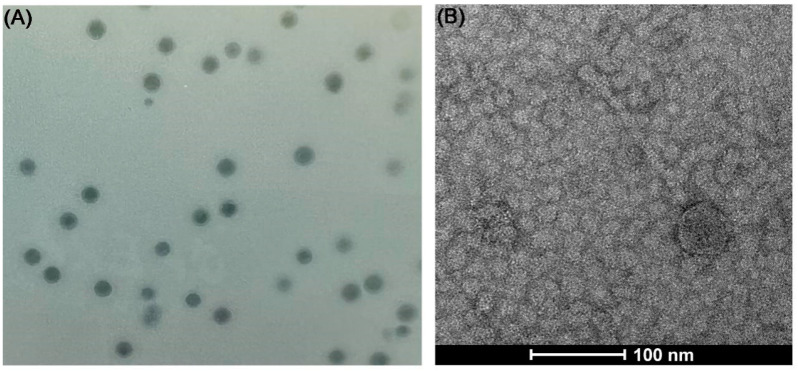
Phage plaques and the morphology and size of phage vB_VnaS-L3. (**A**) Plaques produced by phage vB_VnaS-L3 on the lawn of *V. natriegens* strain AbY-1805. (**B**) Electron micrograph of phage vB_VnaS-L3.

**Figure 3 biology-11-01670-f003:**
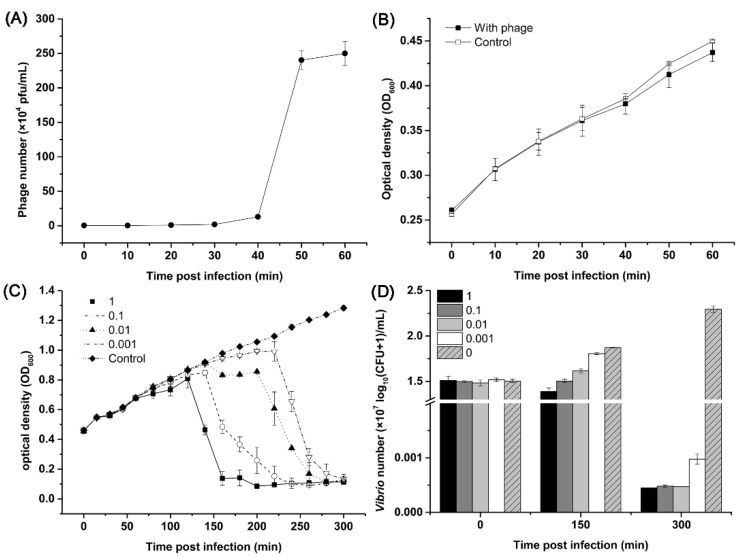
Biological properties of phage vB_VnaS-L3. (**A**) One-step growth curve of phage vB_VnaS-L3. (**B**) The population dynamics of host bacterium *Vibrio natriegens* AbY-1805 with or without phage vB_VnaS-L3 infection. (**C**) Killing curves of *Vibrio natriegens* AbY-1805 by phage vB_VnaS-L3 infection at different MOIs (1, 0.1, 0.01, and 0.001). Error bars represent the standard deviations of triplicate samples. (**D**) *Vibrio* abundance at 0, 150, and 300 min in the lysis assay experiment.

**Figure 4 biology-11-01670-f004:**
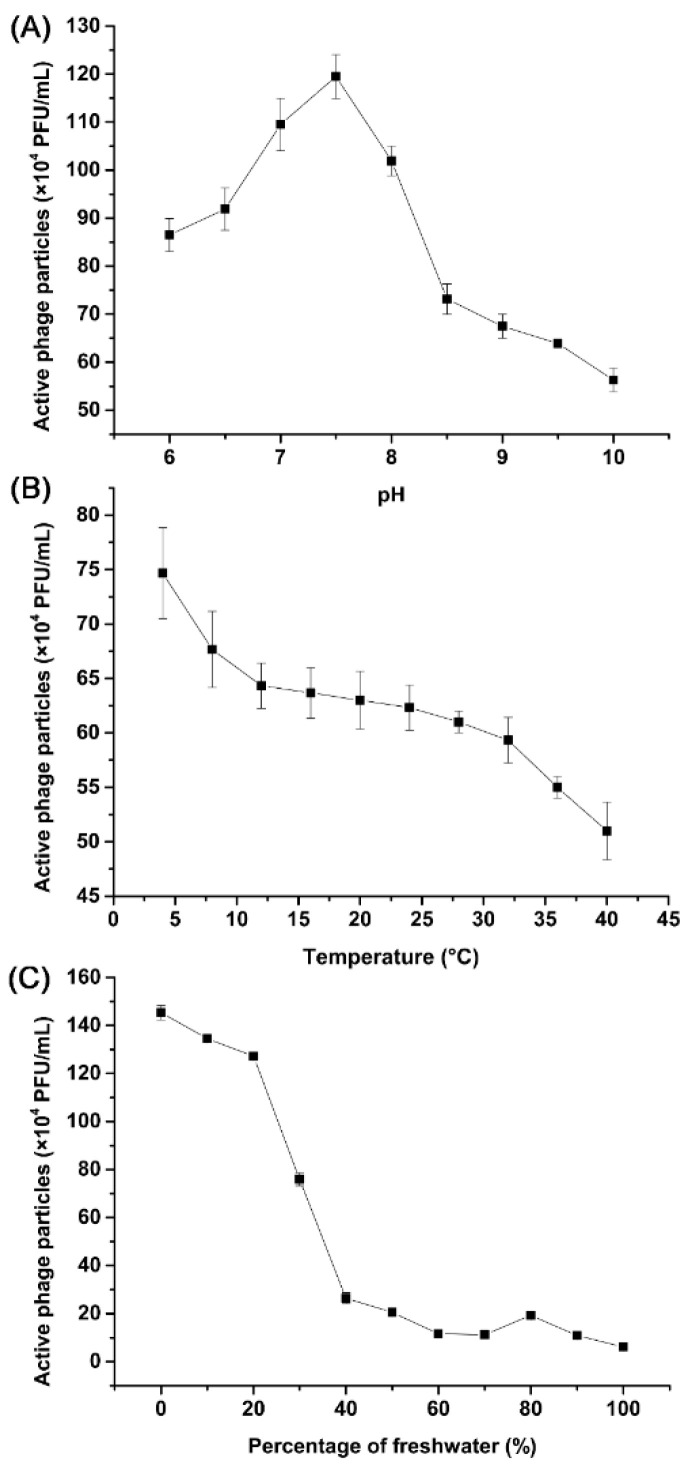
The stability of phage vB_VnaS-L3 under different environmental conditions. (**A**) Stability of phage vB_VnaS-L3 under different pH conditions. (**B**) Stability of phage vB_VnaS-L3 in different temperatures. (**C**) Stability of phage vB_VnaS-L3 under different salinity conditions. Error bars represent the standard deviations of triplicate samples.

**Figure 5 biology-11-01670-f005:**
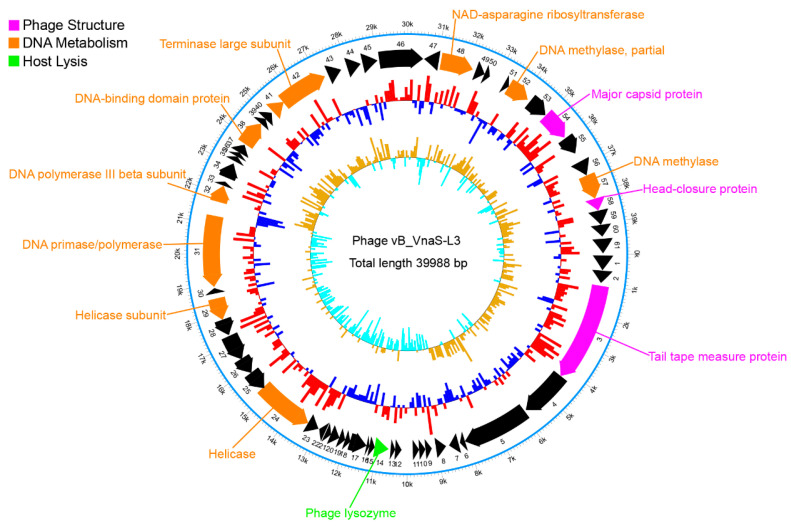
Genomic structure of phage vB_VnaS-L3. The scale representing the genome size (bp) is displayed using the outmost circle. The second outmost circle comprising arrows represents predicted ORFs. The third outmost circle represents the GC contents, with the red color indicating that the GC contents are higher than the average GC content and the blue color indicating that the GC contents are lower than the average GC content. The innermost circle represents the GC-skewness values (GC-skew = (G − C)/(G + C)).

**Figure 6 biology-11-01670-f006:**
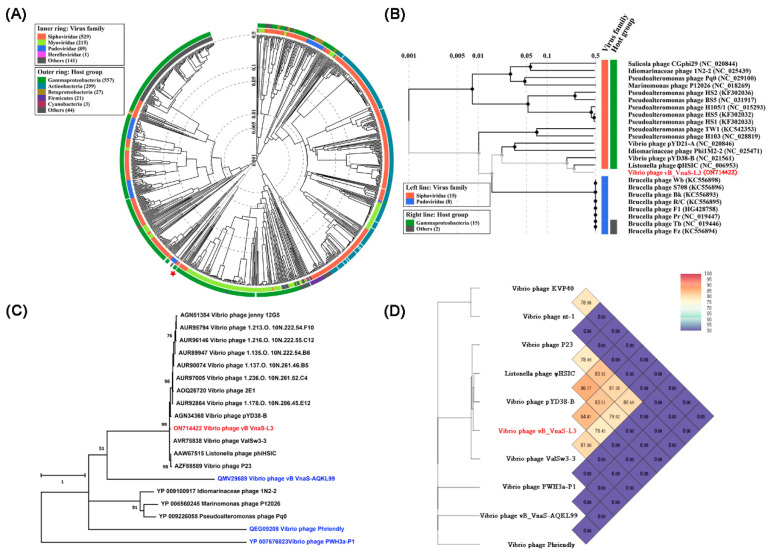
Phylogenetic and comparative genomic analyses of phage vB_VnaS-L3. (**A**) Circular proteomic tree constructed with phage vB_VnaS-L3 genome and similar phage genomic sequences using ViPTree. The colored rings represent viral families (inner ring) and host groups (outer ring). The red star marks the position of phage vB_VnaS-L3 in the phylogenetic tree. (**B**) The viral proteomic tree including phage vB_VnaS-L3 and its 15 nearest phage relatives. The left color bar indicates the viral taxonomic families and the right color bar indicates the host groups. Phage vB_VnaS-L3 is labeled with red color. (**C**) Maximum-likelihood phylogenetic tree constructed using sequences of the viral major capsid proteins. Phage vB_VnaS-L3 is labeled with red color and other *Vibrio natriegens*-infecting phages are labeled with blue color. (**D**) Genome similarity heatmap with OrthoANI values of phage vB_VnaS-L3, its nearest phage relatives, and *V. natriegens*-infecting phages calculated using the OAT software. Phage vB_VnaS-L3 is labeled with red color.

**Figure 7 biology-11-01670-f007:**
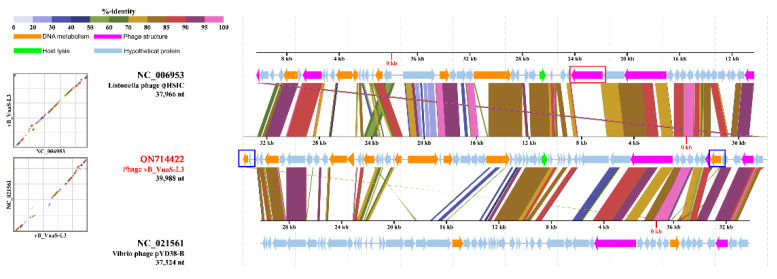
Genome comparisons of phage vB_VnaS-L3, Listonella phage φHSIC, and Vibrio phage pYD38-B. ORFs are depicted by left- or right-oriented arrows depending on the direction of gene transcription. Predicted functional modules are shown with distinct colors. The shaded areas represent sequence similarities at the amino acid level. The box outlined in red marks the functional module that is found only in phage φHSIC, and the boxes outlined in blue mark the functional modules that are found only in phage vB_VnaS-L3.

**Figure 8 biology-11-01670-f008:**
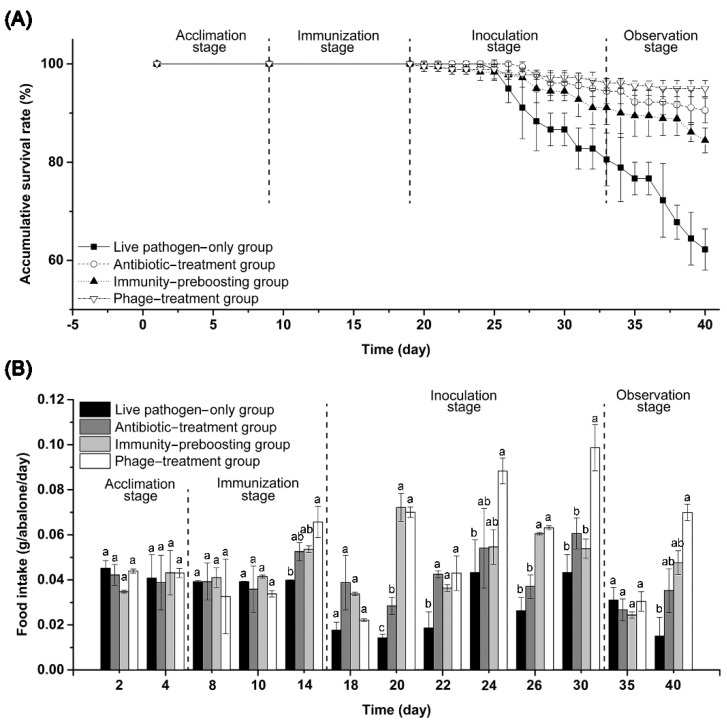
Abalone pathogen challenge bioassay experiment, showing the application potential of phage vB_VnaS-L3 as an effective biocontrol agent against abalone pathogen, *V. natriegens* AbY-1805. (**A**) Accumulative survival rates of abalones, challenged with pathogen *V. natriegens* AbY-1805, under different treatment scenarios. (**B**) The dynamics of abalone feeding rate (g of food intake/abalone) in different treatment groups. Significant differences (ANOVA test, *p* < 0.05) among different groups at each time point are represented by different letters.

**Table 1 biology-11-01670-t001:** Experimental design of antibacterial abalone bioassay.

Groups	Acclimation Stage	Immunization Stage		Inoculation Stage			Observation Stage
	Day 1 to Day 6	Day 7 to Day 16		Day 17 to Day 30			Day 31 to Day 40
	No Treatment	Heat-Inactivated Strain AbY-1805	Phage vB_VnaS-L3	Strain AbY-1805	Phage vB_VnaS-L3	Oxytetracycline	No Treatment
Live pathogen-only group		−	−	+	−	−	
Immunity-preboosting group		+	−	+	−	−	
Antibiotic-treatment group		−	−	+	−	+	
Phage-treatment group		−	+	+	+	−	

“+” or “−” denotes addition or no addition of the specific component in an experiment, respectively.

## Data Availability

The complete genomic sequence of phage vB_VnaS-L3 and the partial 16S rRNA gene sequences of *Vibrio natriegens* strain AbY-1805 and other bacterial strains used in the experiments of this manuscript have been deposited into GenBank under accession numbers ON714422, OP247638, and OP268245-OP268279.

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
