# Peer review of "Isolation and Characterization of a Novel Vibrio natriegens—Infecting Phage and Its Potential Therapeutic Application in Abalone Aquaculture"

_biology, 2022, doi:10.3390/biology11111670_

Round 1
Reviewer 1 Report
In the article “Isolation and characterization of a novel Vibrio natriegens-infecting phage and its potential therapeutic application in abalone aquaculture”, a new phage against a multidrug resistant aquatic pathogen was isolated and characterized. In vivo studies using juvenile Pacific abalones in aquaculture were performed envisioning the use of the phage as a biocontrol agent. The experimental design of the study is good, and the results obtained are of interest for paving the way towards the use of phages for infection control in aquaculture. I would recommend this paper for publication after some revisions have been made.
MAJOR COMMENTS:
1 –Did you test if SM buffer remains stable during the 24 h at the different pHs used? I recommend repeating the test using universal buffer (150 mM KCl, 10 mM KH2PO4, 10 mM Na3C6H5O7, and 10 mM H3BO3), which is more appropriate when testing stability at different pHs.
2 – How did you choose the phage concentration to be used in the phage biocontrol potential bioassay? The killing curves, described on lines 262-263 and represented on figure 3C, are not described in the methods section and should be more discussed. Although the reduction in the OD gives some information about the killing activity of the phage, this assay should be complemented by CFUs determination at some selected time-points. Also, in the end, it would be interesting to select some colonies and test if they are resistant to the phage.
MINOR COMMENTS:
Line 136 – “collocated” should be “collected”
Lines 139-140 – To be easier for the reader to understand the method used, I would say “the double-agar layer method was used, and the presence of phage plaques observed at 24 h, 36 h, and 48 h, respectively”.
Lines 140-141 – How do you collect the phage plaques? Maybe you mean that you “picked” them?
Lines 146-147 – You should test more phage dilutions against the 3 positive bacterial strains since at this concentration you can have a “lysis from without” phenomenon.
Line 447 – “lysogenetic” should be “lysogenic”
Line 476 – Italicize “in vivo”
Reviewer 2 Report
Bacterial diseases affecting aquaculture animals cause large economic losses to producers. In recent years antibiotics and chemotherapeutic agents that are used in aquaculture have been evidenced as loosing efectiveness, resulting in a lack of effective treatment options. As we approach what could be a post-antibiotic era, new prevention and treatment options are being sought. One of such methods is the use of bacteriophages. The authors describe an isolated and characterized bacteriophage against Vibrio natriegens.
The major asset of this manuscript is that it presents a characteristic and information about new lytic phage vB_VnaS-L3. The studies performed on its stability under various environmental conditions are extremely important.
The manuscript is well written. Only minor additions are needed.
Line 123 – please add time of cultivation
Line 139 – please add time of incubation
The bioassay experiment to compare the antibiotic therapy with the effect of phage therapy requires some explanation or new execution with different groups. Please justify lack of the negative control without bacterial challenge and phage treatment, as well as a group in which the safety of the new bacteriophage itself would be checked.
The authors conclude that compared to the group treated with antibiotics, the group treated with phages performed better in increasing survival. Considering that bacteriophages were administered for 24 days, and the antibiotic for 14 days, it is difficult to conclude that. Especially that the authors used phages in only one group, both before and together with the bacteria, which is rather prevention than therapy. The choice of such a division of groups instead of using phages before (like inactivated bacteria) and after infection (like antybiotic) as separate groups requires an additional explanation.
Reviewer 3 Report
This manuscript represents an interesting contribution to the use of phages as an alternative to the use of antibiotics for the control of bacterial pathogens in shellfish aquaculture.
In general, the ms is well written and the scientific content is relevant. However some minor considerations should be taken into account:
- To test the host range of phage vB_VnaS-L3, the authors only used the 3 strains of V. natriengens isolated from abalone in the present study. It would be convenient to include more isolates of this spcies from other origins.
- Table 1 should be moved to the Supplementary material.
- The number of references is excesive (128)!! They must be reduced to those strictly necessary trying to cite, when there are,review references.
Round 2
Reviewer 2 Report
Dear Authors,
I find the improved version of the manuscript satisfactory. I believe that the amendments to the manuscript and the provided explanations are sufficient.
Best regards.